# Estimating Soil Moisture Content in Winter Wheat in Southern Xinjiang by Fusing UAV Texture Feature with Novel Three-Dimensional Texture Indexes

**DOI:** 10.3390/plants14192948

**Published:** 2025-09-23

**Authors:** Tao Sun, Zhijun Li, Zijun Tang, Wei Zhang, Wangyang Li, Zhiying Liu, Jinqi Wu, Shiqi Liu, Youzhen Xiang, Fucang Zhang

**Affiliations:** 1Key Laboratory of Agricultural Soil and Water Engineering in Arid and Semiarid Areas of Ministry of Education, Northwest A&F University, Yangling 712100, China; 2021050986@nwsuaf.edu.cn (T.S.); tangzijun@nwsuaf.edu.cn (Z.T.); 2021012221@nwsuaf.edu.cn (W.Z.); lwy1222@nwsuaf.edu.cn (W.L.); 2024055914@nwsuaf.edu.cn (Z.L.); 2025056093@nwsuaf.edu.cn (J.W.); 2025056042@nwsuaf.edu.cn (S.L.); youzhenxiang@nwsuaf.edu.cn (Y.X.); zhangfc@nwsuaf.edu.cn (F.Z.); 2Xinjiang Research Institute of Agriculture in Arid Areas, Urumqi 830091, China

**Keywords:** winter wheat, multispectral, texture features, correlation matrix, machine learning

## Abstract

Winter wheat is a major staple crop worldwide, and real-time monitoring of soil moisture content (SMC) is critical for yield security. Targeting the monitoring needs under arid conditions in southern Xinjiang, this study proposes a UAV multispectral-based SMC estimation method that constructs novel three-dimensional (3-D) texture indices. Field experiments were conducted over two consecutive growing seasons in Kunyu City, southern Xinjiang, China, with four irrigation and four fertilization levels. High-resolution multispectral imagery was acquired at the jointing stage using a UAV-mounted camera. From the imagery, conventional texture features were extracted, and six two-dimensional (2-D) and four 3-D texture indices were constructed. A correlation matrix approach was used to screen feature combinations significantly associated with SMC. Random forest (RF), partial least squares regression (PLSR), and back-propagation neural networks (BPNN) were then used to develop SMC models for three soil depths (0–20, 20–40, and 40–60 cm). Results showed that estimation accuracy for the shallow layer (0–20 cm) was markedly higher than for the middle and deep layers. Under single-source input, using 3-D texture indices (Combination 3) with RF achieved the best shallow-layer performance (validation R^2^ = 0.827, RMSE = 0.534, MRE = 2.686%). With multi-source fusion inputs (Combination 7: texture features + 2-D texture indices + 3-D texture indices) combined with RF, shallow-layer SMC estimation further improved (R^2^ = 0.890, RMSE = 0.395, MRE = 1.91%). Relative to models using only conventional texture features, fusion increased R^2^ by approximately 11.4%, 11.7%, and 18.1% for the shallow, middle, and deep layers, respectively. The findings indicate that 3-D texture indices (e.g., DTTI), which integrate multi-band texture information, more comprehensively capture canopy spatial structure and are more sensitive to shallow-layer moisture dynamics. Multi-source fusion provides complementary information and substantially enhances model accuracy. The proposed approach offers a new pathway for accurate SMC monitoring in arid croplands and is of practical significance for remote sensing-based moisture estimation and precision irrigation.

## 1. Introduction

Winter wheat (*Triticum aestivum* L.) is a key staple in the arid and semi-arid landscapes of southern Xinjiang, and its growth is highly sensitive to the soil moisture regime [1]. Around the jointing stage, roots extend rapidly, and the demand for water and nutrients rises sharply; when soil moisture content (SMC) is insufficient, photosynthetic rates are suppressed and both vegetative vigor and yield formation fail to reach their potential [2]. Conventional SMC measurements—such as the gravimetric oven-dry method and point-based time-domain reflectometry (TDR)—are accurate but destructive, labor-intensive, and spatially sparse, making them ill-suited to modern agriculture’s need for large-area, real-time, non-destructive monitoring [3].

In recent years, unmanned aerial vehicle (UAV) multispectral remote sensing has shown clear advantages for agricultural water monitoring owing to its high spatial resolution, revisit flexibility, and broad coverage [4,5]. Commercial multispectral cameras typically capture visible (VIS) and near-infrared (NIR) bands concurrently and, with precise spectral calibration and a downwelling light sensor (DLS) [6], provide reliable canopy and soil reflectance for water-stress analysis [7,8]. The NIR (~840 nm) and red-edge (~720/750 nm) bands underpin widely used empirical indices—e.g., the Normalized Difference Water Index (NDWI), Soil-Adjusted Vegetation Index (SAVI), and the Modified Soil-Adjusted Vegetation Index (MSAVI)—which attenuate background effects and enhance water-absorption signals for characterizing surface moisture stress and soil wetness dynamics [9,10]. However, empirical indices built from two or several bands tend to emphasize shallow-layer moisture, and their performance can vary markedly with crop type, region, and phenology; they also struggle to represent the vertical distribution of SMC across soil profiles, which is a key limitation [11].

Texture analysis offers a complementary pathway to mitigate these shortcomings. Gray-level co-occurrence matrix (GLCM)-based metrics (e.g., contrast, entropy, and correlation) quantify local structural heterogeneity of the canopy from a spatial perspective [12]. Under adequate soil moisture, denser canopies and altered leaf morphology/distribution lead to discernible shifts in image texture patterns [13]. Empirical studies have shown that, in scenes with pronounced canopy structural gradients, texture parameters can effectively reflect plant growth status and, when fused with empirical spectral indices, improve SMC prediction; nonetheless, conventional texture features do not capture inter-band interactions across multispectral channels, limiting their accuracy under complex field conditions [14].

To further exploit spectral–spatial coupling, researchers have developed multispectral texture indices using a correlation matrix approach for monitoring crop physiological traits. For example, Yang et al. (2024) [15] combined texture indices with UAV multispectral indices to estimate soybean leaf water content with XGBoost, achieving a coefficient of determination R^2^ = 0.816, root-mean-square error (RMSE) = 1.404, and mean relative error (MRE) = 1.934%. Pei et al. (2024) [16] reported that texture indices constructed from fused UAV multispectral parameters were significantly more sensitive to the crop water stress index (CWSI) than empirical vegetation indices (*p* < 0.05). Extending this line of work, three-dimensional (3-D) hyperspectral indices have been proposed to evaluate soil organic matter [17] and the nitrogen nutrition index [18]; by introducing a third dimension, these indices relax the constraints of traditional 2-D representations. Along similar lines, 3-D texture indices have been used to monitor soil salinity in the Tarim River Basin, yielding markedly higher correlations than conventional vegetation indices and 2-D texture indices [19]. Despite this progress, studies that integrate empirical vegetation indices with multi-dimensional texture indices into a unified dataset for predicting crop physiological indicators remain scarce. Theoretically, 3-D texture indices can capture both joint spectral variation and spatial structural heterogeneity, thereby elucidating canopy spectral responses to differing SMC levels [12]. Yet applications of 3-D texture to field-scale SMC estimation remain limited. Meanwhile, machine-learning algorithms—such as support vector machines (SVM), random forest (RF), and back-propagation neural networks (BPNN)—are well suited to high-dimensional, nonlinear problems and can learn SMC-relevant spectral-texture patterns from multi-feature inputs [20,21]. However, most existing efforts focus on single spectral predictors or shallow-layer SMC, with few providing systematic, mechanistic comparisons of fused multi-source features and machine-learning models across multiple soil depths.

Against this background, the present study leverages UAV multispectral imagery over winter wheat fields in southern Xinjiang, integrating empirical spectral indices with multi-dimensional texture features and using a correlation matrix analysis to select highly responsive predictors. Our objectives are to: (1) quantify the sensitivity of conventional texture features, 2-D texture indices, and 3-D texture indices to SMC across soil layers; and (2) evaluate the performance of machine-learning models coupled with multi-dimensional texture data for estimating SMC at different depths. We anticipate achieving depth-resolved, remote sensing-based monitoring of SMC while advancing the methodological integration of texture information and machine learning, thereby providing a scientific basis and technical support for water-resource management in winter wheat production systems in southern Xinjiang.

## 2. Materials and Methods

### 2.1. Research Area and Experimental Design

The field experiment was conducted at the 224th Regiment of Kunyu City, 14th Division of the Xinjiang Production and Construction Corps (37°14′19″ N, 79°17′34″ E), on the northern foothills of the Kunlun Mountains in the southwestern Tarim Basin. The site lies at 1304–1397 m a.s.l. and has a warm-temperate continental desert climate, with mean annual precipitation of 13.1–48.2 mm, mean annual temperature of 11–12 °C, annual sunshine duration of 2700–2800 h, and a frost-free period of 136–228 d. The soil is sandy; baseline fertility in the 0–15 cm layer was pH = 8.09, soil organic matter = 2.15 g kg^−1^, alkali-hydrolyzable N = 30.78 mg kg^−1^, available *p* = 15.25 mg kg^−1^, and available K = 117.25 mg kg^−1^. Winter wheat cultivar ‘Xindong-44’ was used. Fertilizers included urea (N 46%), diammonium phosphate (DAP; N 18%, P_2_O_5_ 46%), and potassium nitrate (K_2_O 46%, N 13.5%).

A two-factor design was adopted with four drip-irrigation levels—W1 (60% ETc), W2 (80% ETc), W3 (100% ETc), W4 (120% ETc)—and four fertilizer rates (N–P_2_O_5_–K_2_O, kg ha^−1^): F1 (300–280–90), F2 (220–200–70), F3 (140–120–50), and F4 (60–40–30), yielding 16 treatments. Basal fertilizer accounted for 30% of the total and was uniformly broadcast and incorporated into the 0–20 cm layer before the experiment. The remaining fertilizer was applied at regreening, jointing, booting, and grain-filling stages at 20%, 20%, 20%, and 10% of the total, respectively. Each treatment was replicated three times (48 plots in total). Plot dimensions were 20 m × 10 m (200 m^2^). Buffer rows were arranged between adjacent plots and at both ends of the field to minimize edge and treatment-interaction effects.

Wheat was sown on 15 October 2022 and 23 October 2023 at a seed rate of 30 kg mu^−1^ (mu ≈ 666.7 m^2^, 30 kg mu^−1^ ≈ 450 kg ha^−1^) using mechanical planting. Drip-tape spacing was 1 m, with a one-tape–two-row configuration. The fertigation system comprised standard commercial components: pipelines, a 15 L fertilizer tank, a water meter, and 16-mm drip lines with 30 cm emitter spacing. Fertigation followed a 1/4–1/2–1/4 schedule (first quarter with clear water, middle half with fertilized water, final quarter with clear water for flushing). Across the season, 30 irrigation events and seven fertigation events were applied. Crop coefficients (Kc) were set to 0.4 (sowing–seedling), 1.1 (jointing–heading), and 0.6 (grain-filling–harvest) [22].

This study employed a randomized, replicated experimental design in a single, level experimental field with a uniform soil type; all plots used the same cultivar, sowing date, seeding rate, and management practices.

### 2.2. Data Acquisition and Preprocessing

#### 2.2.1. Multispectral Data Acquisition and Processing

UAV flights were conducted at the jointing stage on 20 April 2023 and 17 April 2024, between 11:00 and 13:00 local time under clear skies with stable solar illumination. The flight platform was a DJI Matrice 300 RTK (M300 RTK; SZ DJI Technology Co., Ltd., Shenzhen, China) carrying a Yusense MS600 Pro six-band multispectral camera (Yusense Information Technology and Equipment (Qingdao) Co., Ltd., Qingdao, China). The MS600 Pro is deeply integrated with the M300 via the DJI X-Port interface and supports a downwelling light sensor (DLS; Yusense, Qingdao, China) together with white-panel calibration (Spectralon^®^ diffuse reflectance standard, Labsphere, North Sutton, NH, USA) to retrieve surface reflectance; all six multispectral channels use 12-bit global-shutter, synchronized imaging. Typical sensor specifications include a 49.5° × 38.1° field of view and an approximate ground sampling distance (GSD) of 8.65 cm pixel^−1^ at 120 m altitude. The standard bands and full widths at half maximum (FWHM) are: blue 450 nm @ 30 nm, green 555 nm @ 27 nm, red 660 nm @ 22 nm, red-edge 720 nm @ 10 nm, red-edge 750 nm @ 10 nm, and near-infrared (NIR) 840 nm @ 30 nm. These bands target crop pigments and canopy structure: the red band captures chlorophyll absorption; the red-edge bands are sensitive to chlorophyll and structural changes; NIR relates to canopy structure and water status; and blue/green complement pigment and backscatter responses. Detailed sensor specifications are provided in [15].

Flight lines were planned to cover the experimental area, and a certified white reference panel was measured immediately before each mission for reflectance calibration. Flights were conducted at 30 m above ground level, yielding an effective GSD of approximately 2.2 cm pixel^−1^ with this optical system—balancing the need to resolve canopy structural elements with efficient area coverage and stable radiometry, consistent with recommendations to match pixel size to characteristic canopy features for texture analysis. Airspeed was maintained at 2.5 m s^−1^, with forward and side overlaps of 75% and 65%, respectively. To ensure geometric and radiometric consistency, overlap settings and airspeed were kept identical across missions.

Image mosaicking, geometric correction, and radiometric correction were performed in YuSense Map v2.2.2 (Yusense Information Technology and Equipment (Qingdao) Co., Ltd., Qingdao, China). The preprocessed mosaics were imported into ENVI 5.3 (Harris Geospatial Solutions, Broomfield, CO, USA) to compute eight gray-level co-occurrence matrix (GLCM)-based texture features (TFs) using second-order statistics: mean (MEA), variance (VAR), homogeneity (HOM), contrast (CON), dissimilarity (DIS), entropy (ENT), second moment (SEC), and correlation (COR) [16]. GLCM texture features were computed on reflectance mosaics using a 5 × 5 moving window and a one-pixel offset at 0°, 45°, 90°, and 135°, with directional results averaged to reduce rotational effects. At the 30 m flight altitude, this window spans approximately 11 cm, matching the spatial scale of wheat leaves and stem communities. To mitigate soil–vegetation mixing, we generated a vegetation mask from an NIR-based index and retained pixels with index values > τ (τ selected from the image histogram within 0.30–0.40). Small gaps and shadows were refined using 3 × 3 morphological opening and closing, and the mask was applied prior to computing GLCM textures and multiband indices. All imagery was analyzed in the reflectance domain: for each mission, a white reference panel was photographed and a downwelling light sensor (DLS) recorded incident irradiance; radiometric calibration followed a consistent workflow to produce reflectance mosaics, after which texture features (GLCM, etc.) were calculated [23]. For cross-flight comparability, gray-level quantization was fixed at 64 levels, the window size at 5 × 5 pixels, and the offset at one pixel with the four directions averaged; these parameters were held constant across all dates and plots.

The corresponding formulas are given in Equations (1)–(8).(1)$$MEA=\sum_{i,\;j=1}^{G}\left(iP(i,\;j)\right)$$(2)$$VAR=\sum_{i=1}^{G}\sum_{j=1}^{G}{\left(i-u\right)}^{2}P(i,\;j)$$(3)$$HOM=\sum_{i=1}^{G}\sum_{j=1}^{G}\frac{P(i,\;j)}{1+{\left(i-j\right)}^{2}}$$(4)$$CON=\sum_{i=1}^{G}\sum_{j=1}^{G}{\left(i-j\right)}^{2}P(i,\;j)$$(5)$$DIS=\sum_{i=1}^{G}\sum_{j=1}^{G}P(i,\;j)\left|i-j\right|$$(6)$$ENT=\sum_{i=1}^{G}\sum_{j=1}^{G}P(i,\;j)\log P(i,\;j)$$(7)$$SEC=\sum_{i=1}^{G}\sum_{j=1}^{G}P^{2}(i,\;j)$$(8)$$COR=\sum_{i=1}^{G}\sum_{j=1}^{G}\frac{\left(i-MEA_{j})(j-MEA_{i})P(i,\;j\right)}{\sqrt{VAR_{i}}\sqrt{VAR_{j}}}$$

Building on prior studies, six two-band texture indices were constructed. Feature screening followed a correlation matrix approach. We computed Pearson correlations between SMC and each of the 48 band–texture “positions” (8 GLCM metrics × 6 spectral bands), as well as their derived combinations. Features with *p* < 0.05 were retained. To control multicollinearity among significant features, we removed pairs with inter-feature correlation |r| ≥ 0.90, yielding a nonredundant subset for modeling. Here, Band1–Band6 correspond to blue, green, red, red-edge1, red-edge2, and NIR, respectively; for example, “HOM6” denotes GLCM homogeneity computed from Band6 (NIR) additive texture index (ATI), ratio texture index (RTI), normalized difference texture index (NDTI), difference texture index (DTI), reciprocal difference texture index (RDTI), and reciprocal additive texture index (RATI)—to explore the utility of texture information for UAV-based SMC estimation in winter wheat. Detailed definitions are provided in Table 1. Here, *T_i_* and *T_j_* denote (arbitrarily selected) texture features.

In addition, four three-dimensional (3-D) texture indices were proposed (Table 2):

#### 2.2.2. Soil Moisture Measurement

Immediately after UAV imaging, soil samples were collected with an auger at nine random points per plot to a depth of 0–60 cm and stratified into three layers: 0–20, 20–40, and 40–60 cm [24]. Gravimetric SMC was determined by oven-drying at 105 °C (BINDER GmbH, Tuttlingen, Germany) for ≥8 h, with masses measured before and after drying on a balance (precision 0.001 g, OHAUS Corporation, Parsippany, NJ, USA). Gravimetric SMC was converted to volumetric SMC (m^3^ m^−3^) using bulk density. Plot-level SMC was taken as the mean of the nine sampling points.

### 2.3. Modeling Approach

To achieve accurate SMC estimation, a three-stage modeling pipeline was implemented. First, univariate correlation tests (*p* < 0.05) were used to select spectral/texture predictors significantly associated with SMC. Second, three machine-learning models—partial least squares regression (PLSR) [25], back-propagation neural network (BPNN) [26], and random forest (RF) [27]—were built to assess algorithmic performance for complex nonlinear fitting. Third, model hyperparameters were tuned via cross-validation and grid search to identify optimal configurations and enhance accuracy and stability. Additionally, for each soil depth, candidate features underwent a two-stage screening: (i) correlation screening (Pearson’s correlation with SMC; *p* < 0.05); and (ii) multicollinearity control, in which feature pairs with |r| > 0.90 were removed and variance inflation factors were required to satisfy VIF ≤ 10 (VIFj = 1/(1-Rj^2^)). The retained subset was then used for model fitting.

Modeling employed a depth-stratified train/validation split (approximately 2/3: 1/3). Within the training set, 10-fold cross-validation (folds stratified by depth) was used for hyperparameter selection and to prevent information leakage. All predictors were z-standardized (mean 0, variance 1) using statistics computed on each training fold; these parameters were subsequently applied to the corresponding validation fold and to the held-out validation set. Parameter settings were as follows. RF used the Gini criterion, with 200 trees and max features = “auto”; out-of-bag (OOB) error was used to assess stability, and convergence was checked over multiple training iterations [28]. For PLSR, ten-fold cross-validation determined the optimal number of latent variables (LVs); to avoid overfitting, the number of LVs was chosen as the smallest value achieving a ≥ 5% incremental increase in the cumulative explained variance of Y [18]. The BPNN adopted a single hidden layer with hyperbolic tangent (TANSIG) activation and Levenberg–Marquardt (trainLM) optimization. The number of hidden neurons was increased in steps of five from 10 to 100, and validation performance was used to select the optimum; 45 neurons yielded the best convergence behavior [21]. All settings were held constant across depths and applied under identical preprocessing and train/validation splits to ensure fair comparison.

### 2.4. Dataset Partitioning and Model Evaluation

At the jointing stage, SMC samples were obtained for 96 plots across three soil layers (*n* = 96). Data were split by stratified random sampling into a modeling set (two-thirds of the samples) and a validation set (one-third). Descriptive statistics for each soil layer are shown in Figure 1. Model performance was quantified using the coefficient of determination (R^2^), root-mean-square error (RMSE), and mean relative error (MRE; also termed mean relative deviation) [24,29,30].

## 3. Results and Analysis

### 3.1. Correlation Between Multispectral Texture Parameters and Soil Moisture Content in Winter Wheat

The correlation coefficients between multispectral texture parameters and SMC are summarized in Table 3, Table 4 and Table 5 and Figure 2 and Figure 3. Overall, for a given texture parameter, its correlation with the shallow layer (0–20 cm) was consistently higher than that with the middle (20–40 cm) and deep (40–60 cm) layers. Specifically, the texture features showing the highest correlations with the shallow, middle, and deep layers were DIS of Band 1, VAR of Band 4, and VAR of Band 3, with coefficients of 0.622, 0.612, and 0.576, respectively. The top-performing two-dimensional texture indices were ATI (HOM2 and MEA2) for the shallow layer, ATI (HOM2 and MEA2) for the middle layer, and DTI (MEA3 and MEA4) for the deep layer, with correlation coefficients of 0.643, 0.615, and 0.583, respectively. Among the three-dimensional texture indices, the highest correlations were achieved by DTTI (HOM5, HOM2, and SEC6) for the shallow layer, MSI (COR4, VAR6, VAR4) for the middle layer, and DTTI (HOM4, DIS1, and HOM4) for the deep layer, with coefficients of 0.644, 0.632, and 0.621, respectively.

At a fixed soil depth, the three-dimensional texture indices provided the strongest associations with SMC. In particular, for the shallow layer (0–20 cm), DTTI (HOM5, HOM2, SEC6) yielded the highest correlation (r = 0.644).

### 3.2. Model Construction for SMC at Different Depths

Based on the significant correlations identified above, we assembled different model input combinations from (i) texture features, (ii) two-dimensional texture indices, and (iii) three-dimensional texture indices, and constructed SMC estimation models at three depths using RF, PLSR, and BPNN. Combination 1 comprised significantly correlated texture features; Combination 2, significantly correlated 2-D texture indices; Combination 3, significantly correlated 3-D texture indices; Combination 4, the union of Combinations 1 and 2; Combination 5, the union of Combinations 1 and 3; Combination 6, the union of Combinations 2 and 3; and Combination 7, the union of Combinations 1–3. Modeling results are shown in Figure 4 and Figure 5, which are based on Figure 5’s highlights, the “classical” input (Combination 1) and the best-performing fused input (Combination 7).

Results showed that, for a given input combination and learning algorithm, the shallow-layer (0–20 cm) models consistently achieved higher accuracy than those for the middle and deep layers. Focusing on the shallow layer, the best single-source input was Combination 3 (3-D texture indices) paired with RF, yielding a validation R^2^ of 0.827, RMSE of 0.534, and MRE of 2.686%. Under multi-source inputs (Combinations 4–7), the highest accuracy was obtained with Combination 7 (i.e., texture features + 2-D texture indices + 3-D texture indices) using RF, with R^2^ = 0.890, RMSE = 0.395, and MRE = 1.910%.

Notably, relative to the traditional input (Combination 1: texture features) under the same RF algorithm, the fused input (Combination 7) increased validation R^2^ by approximately 11.4%, 11.7%, and 18.1% for the shallow, middle, and deep layers, respectively.

## 4. Discussion

Accurate determination of SMC is fundamental to yield improvement and water management in modern agriculture [31]. Ground-based measurements are labor-intensive, time-consuming, and often lack spatial representativeness, whereas satellite remote sensing—despite its synoptic coverage—is constrained by cloud contamination and struggles to meet the simultaneous demands for high spatial and temporal resolution at field scales [32]. In contrast, unmanned aerial vehicle (UAV) multispectral remote sensing can rapidly acquire high-resolution vegetation observations with short turnaround time, low cost, and operational flexibility, thereby complementing satellite data for agricultural fields [33]. Vegetation indices and texture information derived from UAV imagery indirectly reflect canopy growth and water status and thus provide a basis for SMC estimation [34,35]. Consequently, integrating UAV-based multispectral vegetation indices with novel texture features is technically feasible and holds strong application potential for SMC monitoring.

In this study, SMC estimation accuracy was markedly higher for the shallow layer (0–20 cm) than for deeper layers. This primarily reflects the sensing geometry and crop physiology: UAV multispectral imagery characterizes the canopy, while the effective water uptake zone of winter wheat roots is concentrated in the shallow soil [36,37]. Fluctuations in shallow-layer moisture promptly modulate plant water status and leaf water content, rendering canopy-derived indices highly sensitive to near-surface SMC [38]. By contrast, moisture changes at greater depths induce weaker canopy spectral responses, leading to lower correlations between deep-layer SMC and reflectance [26]. The texture parameters observed by UAV primarily encode canopy water status; hence, they correlate most strongly with shallow-layer SMC, translating into superior shallow-layer model performance. The superiority of three-dimensional (3-D) texture indices over conventional two-dimensional (2-D) texture metrics stems from their fusion of multi-band texture information and expansion of the feature space through correlation/covariance formulations. Prior work indicates that 3-D texture indices, constructed via a correlation matrix framework, can capture richer spatial-structural information and more comprehensively represent canopy complexity [19]. In our case, the DTTI index—formed from homogeneity and second-moment features across different bands—achieved the highest correlation for the shallow layer, indicating effective fusion of spectral and structural cues. Specifically, homogeneity (HOM) and second moment (SEC) in DTTI quantify reflectance uniformity and local contrast, both of which are highly sensitive to water stress [39]. Compared with single-band texture features, 3-D texture indices couple spectral and spatial dimensions, enabling detection of subtle SMC-driven differences while accounting for canopy spatial organization.

Regarding multi-input fusion, jointly using raw texture features, 2-D texture indices, and 3-D texture indices substantially improved SMC estimation accuracy. This multi-source integration provides complementary information: raw texture features capture micro-structural changes in the canopy; 2-D texture indices emphasize water-sensitive contrasts via band combinations [15]; and 3-D texture indices capture more complex spatial-textural patterns through high-dimensional coupling [19]. Empirical evidence supports these gains—for instance, in soil salinity retrieval, a spectral–texture–2-D–3-D (SOTT) feature set increased the RF model’s R^2^ from 0.76 to 0.90 [19]. Analogously, our best-performing Combination 7 (texture features + 2-D + 3-D texture indices) achieved maximal sensitivity to SMC variation in winter wheat fields in southern Xinjiang. Recent UAV-based studies indicate that, across crops and land-cover types, spectral-only models provide a reasonable baseline for retrieving soil moisture content (SMC) or diagnosing water stress, but augmenting them with gray-level co-occurrence matrix (GLCM)-based texture features and/or thermal signals consistently improves accuracy—for example, in winter wheat (water-related traits and leaf area index, where VIs + texture > VIs alone) [40], soybean SMC (fusion of VIs + texture indices + thermal infrared, TIR, performed best) [24], and drip-irrigated citrus orchards (RGB + multispectral + thermal fusion outperformed single-source inputs) [34]. Depth-wise patterns are likewise consistent with our findings: canopy-driven multispectral and texture features are most sensitive to shallow soil layers (0–20 cm) [24], whereas incorporating thermal signals enables accurate mapping of crop SMC across multiple depths. In our results, the models exhibit a mild overestimation at the dry end and underestimation at the wet end. This end-member bias likely reflects regression-to-the-mean under a limited dynamic range and partial saturation of canopy texture/spectral responses at moisture extremes [41], potentially compounded by class imbalance at very low/high SMC.

By leveraging multi-dimensional spatial information, it more fully reflected soil moisture conditions and significantly enhanced predictive skill. Random forest (RF) outperformed partial least squares regression (PLSR) and back-propagation neural networks (BPNN), largely due to its ensemble architecture and learning strategy [42]. RF trains multiple decision trees on bootstrapped samples and randomized feature subsets, conferring strong nonlinear fitting capacity and generalization [43]. The injected randomness mitigates overfitting and maintains stability in high-dimensional settings; in addition, RF provides variable-importance measures that inform feature selection [44]. In contrast, PLSR relies on linear projections and is limited in capturing complex nonlinearities [3], while BPNN, although capable of nonlinear modeling, operates as a black box, is sensitive to hyperparameters, can become trapped in local minima, and typically requires larger datasets [45]. Overall, RF’s parallel tree structure and ensemble strategy yield higher accuracy and robustness when handling multi-dimensional, correlated remote-sensing features [46,47]. Mechanistically, these observations are consistent with the data regime of this study (moderate sample size, multi-depth targets, and correlated texture/index predictors). RF’s bagging reduces variance, and the random subspace step decorrelates trees, which is advantageous when predictors exhibit multicollinearity and mixed relevance [48,49]. The tree-based splits naturally encode interactions among bands and texture families (GLCM, 2-D, 3-D indices) without requiring explicit feature engineering.

Moreover, out-of-bag estimates offer near-cross-validation diagnostics, and permutation importance pinpoints the most informative “positions,” supporting parsimonious modeling and interpretability at the feature-family level. Limitations of RF—piecewise-constant responses and weaker extrapolation at distribution tails—are mitigated here by the depth-specific training and the use of normalized error metrics, so performance gains primarily reflect better capture of nonlinear structure rather than overfitting. PLSR, by contrast, constructs orthogonal latent variables that maximize covariance between predictors and response, offering a low-variance linear baseline and inherent tolerance to collinearity [50]. This is valuable for small-sample, high-*p* settings; however, when canopy–soil radiative responses are nonlinear (e.g., band-dependent saturation, interaction between red-edge/NIR structure and texture contrast), a linear latent-space mapping underfits, which explains its consistently lower skill relative to RF despite its stability and interpretability. PLSR still serves as a transparent benchmark defining the linear “ceiling” for our feature set [51]. BPNN provides universal function approximation and, with a TANSIG activation and Levenberg–Marquardt optimization, converges rapidly in well-conditioned problems [52]. Yet under limited sample sizes and noisy, partially collinear inputs, BPNN is sensitive to hidden-size choice, weight initialization, and regularization strength; it may settle at suboptimal local minima and exhibit variance across runs unless strong early-stopping/weight decay is enforced [53]. In our setting, the search over 10–100 hidden neurons identified an optimum that performed credibly but not as robustly as RF, consistent with BPNN’s greater data demand for stable generalization. Taken together, the ranking RF > PLSR ≈ BPNN observed here is therefore mechanistically coherent: RF best matches the problem structure (nonlinear interactions + correlated predictors + moderate *N*), PLSR offers a strong linear baseline under collinearity, and BPNN’s capacity is not fully realized at the current data scale. These results support our conclusion that fusing multi-dimensional texture information is beneficial, and that RF is a pragmatic, accurate, and robust choice for depth-resolved SMC estimation under the present experimental conditions.

This study is exploratory and focuses on field-scale conditions for winter wheat in the arid region of southern Xinjiang at the jointing/tillering stage; therefore, extrapolation is constrained to this crop–growth-stage combination. Extending the findings to other crops or varieties, different phenological stages, or alternative sites requires reapplying the workflow used here: (i) harmonized data acquisition and preprocessing; (ii) construction of candidate 2D/3D texture indices; (iii) correlation and collinearity screening; and (iv) modeling and independent validation. In doing so, sensitive “position” combinations should be reselected, and the models retrained and revalidated. Ideally, multi-temporal sampling would be incorporated to develop stage-specific or cross-stage models, thereby improving temporal robustness and transferability. To address reviewers’ requests for comparisons with prior studies, we note that our multi-source texture-fusion concept is consistent with the direction of existing UAV research (spectral only; spectral plus 2D texture; different surface types), and the accuracy gains we obtained for jointing-stage wheat are comparable to, or better than, commonly reported results—further supporting the effectiveness and scalability of 3D texture fusion. At the sensor level, current multispectral platforms are limited by the number of bands and by spatial resolution (our system provides only six VIS–NIR bands and lacks SWIR; operating altitude also constrains ground sample distance, GSD), which creates a bottleneck for detecting subtle differences in soil moisture content (SMC) [54,55,56,57].

Future work could employ hyperspectral or multi-angle and higher-resolution imagery and explore fusion with complementary sensing modalities such as thermal infrared and radar. At the modeling level, mainstream machine-learning methods still fall short in interpretability; introducing explainable learning or physics–data hybrid frameworks (differentiable physics plus ML) could enhance physical consistency and credibility [58,59]. In addition, the limited sample size, sparse observation times, and incomplete phenological coverage may affect temporal stability and cross-environment generalization. We therefore recommend expanding to multi-site, multi-year, and multi-stage datasets and employing strategies such as leave-one-site-out validation and time-blocked cross-validation to obtain more reliable extrapolation performance in high-dimensional feature spaces [60].

It should also be emphasized that species or varietal differences alter leaf-angle distribution, plant architecture, and canopy closure, thereby influencing texture expression and the ranking of the most sensitive combinations. Non-water stresses such as nutrient deficiency and diseases may likewise trigger texture changes that do not directly correspond to SMC; accordingly, it is advisable to record or incorporate relevant stress covariates (e.g., chlorophyll or thermal indicators, pest and disease records) to distinguish water-related from non-water effects [61,62]. Meanwhile, texture features evolve with phenology (e.g., changes in between-row bare-soil fraction, LAI, and structural reorganization). Stage-specific or multi-task modeling combined with phenological covariates (e.g., growing degree days, stage encoding) or temporal normalization is recommended to achieve generalization across time and growth stages [63,64]. Overall, the 3D texture combinations prioritized in this study are most suitable for jointing-stage winter wheat; when transferring to other crops, stages, or sites, the same screening–modeling–validation pipeline should be followed to reidentify sensitive combinations and to complete independent validation, ensuring optimal performance and robustness.

## 5. Conclusions

Under comparable inputs, shallow-layer (0–20 cm) SMC was estimated with substantially higher accuracy than middle layers, consistent with the shallow effective water-uptake zone in winter wheat and the canopy’s heightened spectral sensitivity to near-surface moisture. The proposed 3-D texture indices—especially DTTI—exhibited the strongest response to shallow-layer SMC (r = 0.644), clearly surpassing single-band conventional texture features. Among input combinations, the single-source 3-D texture indices (Combination 3) paired with RF already achieved high accuracy for the shallow layer (validation R^2^ = 0.827). When raw texture features, 2-D texture indices, and 3-D texture indices were fused (Combination 7), the RF model attained the best shallow-layer performance (R^2^ = 0.890; RMSE = 0.395; MRE = 1.91%). Relative to using only conventional texture features, the fused Combination 7 increased R^2^ by approximately 11.4%, 11.7%, and 18.1% for the shallow, middle, and deep layers, respectively. In summary, a multi-source framework that integrates UAV multispectral imagery, multi-dimensional texture indices, and machine-learning models confers clear advantages for estimating shallow-layer SMC in croplands and offers reliable technical support for precision irrigation and water management in arid winter wheat systems.

## Figures and Tables

**Figure 1 plants-14-02948-f001:**
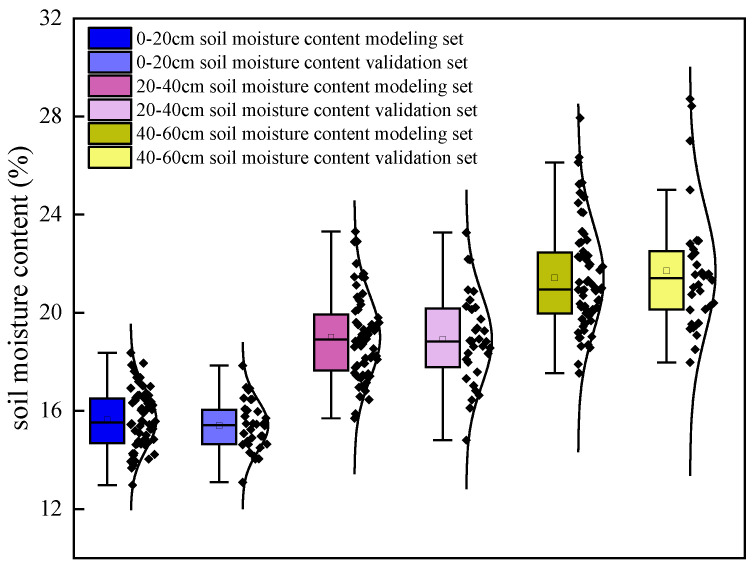
Statistical characteristics of soil moisture content modeling and validation sets for winter wheat in different soil layers.

**Figure 2 plants-14-02948-f002:**
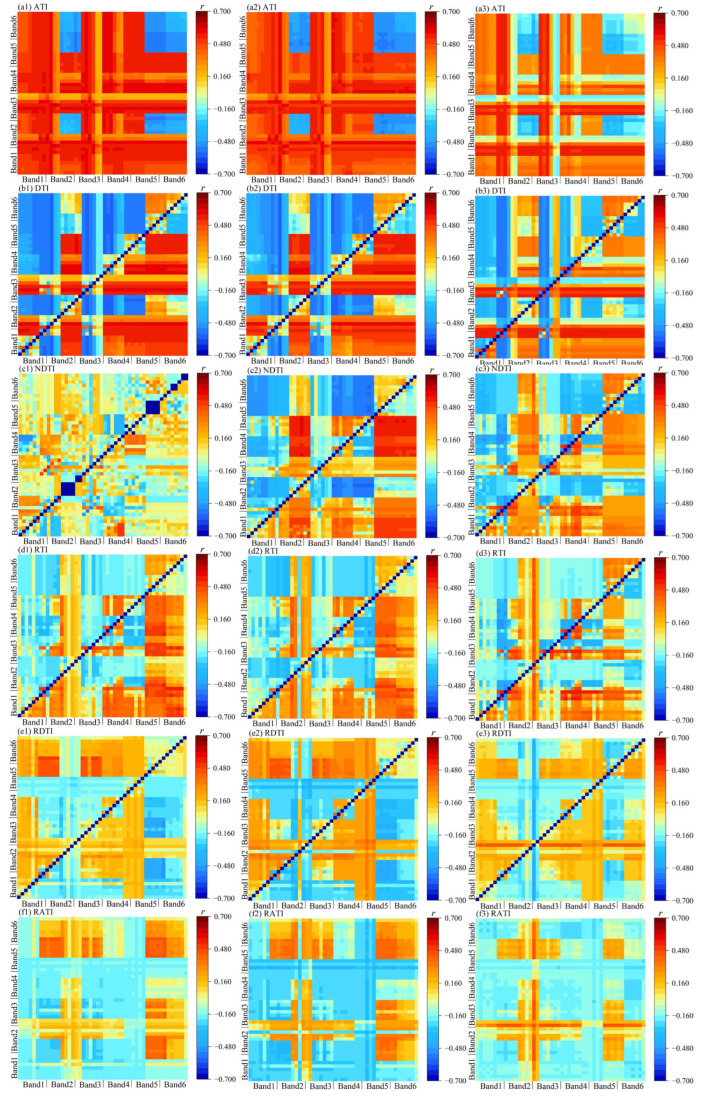
Correlation coefficients between soil moisture content and the 2-D texture index for each winter wheat soil layer. Each cell shows the Pearson correlation (r) between SMC and a 2-D texture index formed by two “positions” (texture × band, e.g., HOM6, MEA2). The color bar denotes r (−1 to 1). “Position” notation: fᵦ denotes texture f at band b; example: RTI (HOM6, MEA2).

**Figure 3 plants-14-02948-f003:**
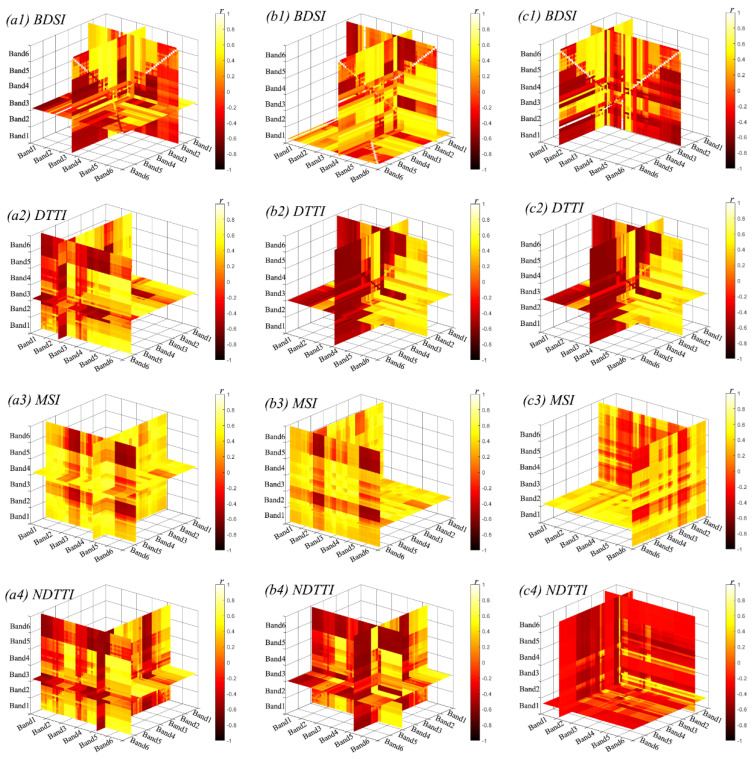
Correlation coefficients between soil moisture content and the 3-D texture index for each winter wheat soil layer. Each cell shows the Pearson correlation (r) between SMC and a 3-D texture index formed by three “positions” (texture × band). The color bar denotes r (−1 to 1); darker colors indicate larger |r|. Example: BDSI (HOM6, MEA2, and VAR3).

**Figure 4 plants-14-02948-f004:**
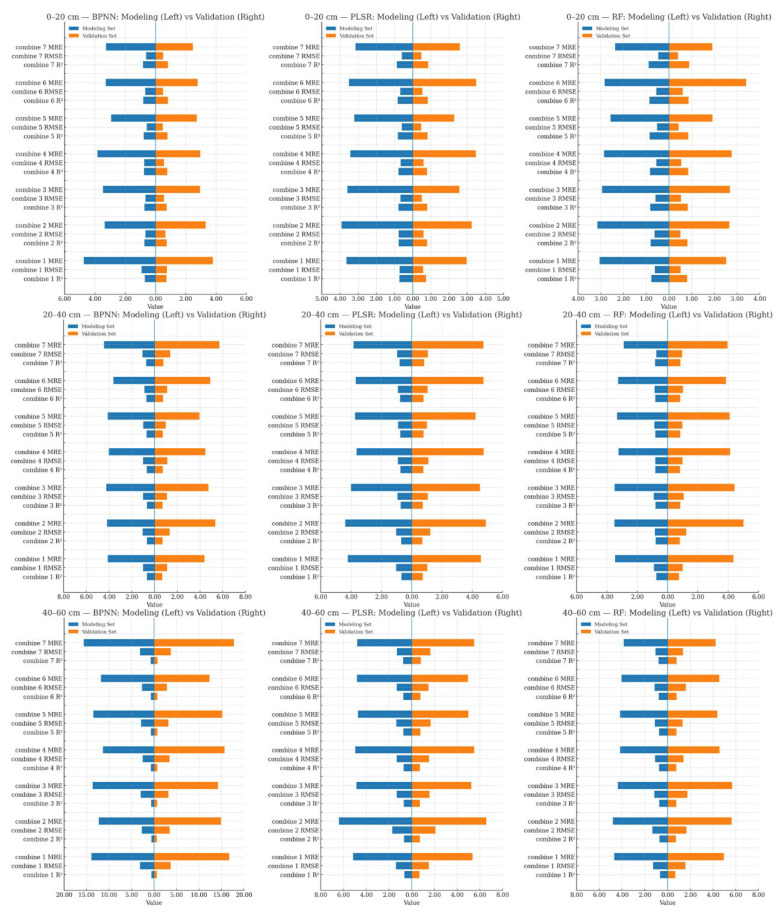
Comparison of the accuracy test results of the estimation model validation set. To preserve visual resolution under depth-specific value ranges, panels are plotted with independent axis limits (R^2^ is comparable, whereas RMSE/MRE vary in magnitude). Cross-depth comparisons should rely on normalized metrics and slope/intercept rather than raw axis scales. Axis choices are for visualization only and do not affect the reported statistics.

**Figure 5 plants-14-02948-f005:**
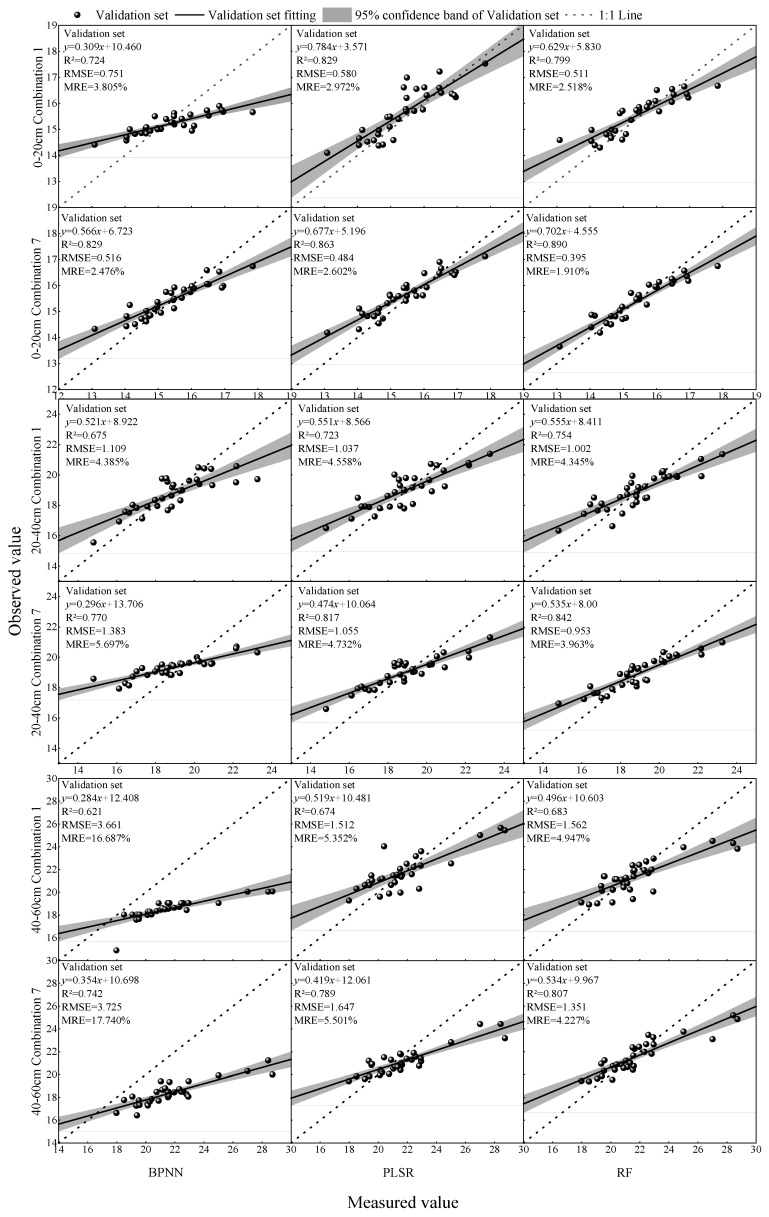
The input variables were texture features (traditional input, combination 1) and a multi-source remote sensing dataset (optimal input combination 7 in Figure 4: texture features + 2-D texture index + 3-D texture index). The prediction results of the validation sets for soil moisture estimation models for winter wheat in different soil layers constructed using different machine learning models were compared. Points show validation samples only; the solid line is the least-squares fit, and the dashed line is the 1:1 reference. Reported R^2^, RMSE, and MRE refer to the validation set.

**Table 1 plants-14-02948-t001:** Texture index calculation formula.

Texture Indices	Formula	References
ATI	*ATI* = $T_{i}+T_{j}$	[12]
DTI	*DTI* = $T_{i}-T_{j}$	[12]
NDTI	*NDTI* = $\left(T_{i}-T_{j})/(T_{i}+T_{j}\right)$	[12]
RTI	*RTI* = $T_{i}/T_{j}$	[15]
RDTI	*RDTI* = ($1/T_{i})-(1/T_{j})$	[15]
RATI	*RATI* = $\left(1/T_{i})+(1/T_{j}\right)$	[15]

Note: *T_i_* and *T_j_* denote selected GLCM texture metrics computed at bands *i* and *j,* respectively (e.g., HOM_6_ = homogeneity at Band 6). Bands 1–6 map to Blue, Green, Red, Red-edge1, Red-edge2, and NIR. All textures are unitless and computed on reflectance mosaics with fixed parameters.

**Table 2 plants-14-02948-t002:** Three-dimensional texture index calculation formula.

Texture Indices	Formula
BDSI	*BDSI* = $\left(T_{i}-T_{j})/(T_{k}-T_{j}\right)$
DTTI	*DTTI* = $T_{i}-T_{j}-T_{k}$
MSI	*MSI* = $\left(T_{i}\times T_{j}\times T_{k}\right)$
NDTTI	*NDTTI* = $\left(T_{i}-T_{j}-T_{k})/(T_{i}+T_{j}+T_{k}\right)$

Note: *T_i_*, *T_j_* and *T_k_* denote selected GLCM texture metrics computed at bands *i, j* and *k*, respectively (e.g., HOM_6_ = homogeneity at Band 6). Bands 1–6 map to Blue, Green, Red, Red-edge1, Red-edge2, and NIR. All textures are unitless and computed on reflectance mosaics with fixed parameters.

**Table 3 plants-14-02948-t003:** Correlation coefficient analysis between texture characteristics and soil moisture content in different soil layers.

Soil Depth	Texture Feature	Correlation Coefficient (*p* < 0.05)					
		Band 1	Band 2	Band 3	Band 4	Band 5	Band 6
0–20 cm	MEA	0.485 *	0.478 *	0.481 *	0.493 *	0.539 *	0.543 *
	VAR	0.568 *	0.585 *	0.558 *	0.609 *	0.371 *	0.304 *
	HOM	0.350 *	0.267 *	0.311 *	0.316 *	0.442 *	0.421 *
	CON	0.585 *	0.601 *	0.567 *	0.498 *	0.211 *	0.173
	DIS	0.622 *	0.593 *	0.619 *	0.488 *	0.336 *	0.306 *
	ENT	0.574 *	0.581 *	0.572 *	0.583 *	0.577 *	0.578 *
	SEC	0.559 *	0.571 *	0.563 *	0.563 *	0.565 *	0.577 *
	COR	0.377 *	0.388 *	0.377 *	0.363 *	0.317 *	0.320 *
20–40 cm	MEA	0.421 *	0.413 *	0.407 *	0.434 *	0.479 *	0.489 *
	VAR	0.517 *	0.539 *	0.483 *	0.612 *	0.414 *	0.353 *
	HOM	0.396 *	0.335 *	0.356 *	0.417 *	0.404 *	0.456 *
	CON	0.538 *	0.571 *	0.492 *	0.548 *	0.279 *	0.242 *
	DIS	0.593 *	0.584 *	0.562 *	0.521 *	0.387 *	0.358 *
	ENT	0.533 *	0.532 *	0.525 *	0.535 *	0.534 *	0.537 *
	SEC	0.527 *	0.536 *	0.542 *	0.534 *	0.523 *	0.532 *
	COR	0.439 *	0.423 *	0.440 *	0.385 *	0.382 *	0.372 *
40–60 cm	MEA	0.389 *	0.362 *	0.403 *	0.338 *	0.288 *	0.272 *
	VAR	0.570 *	0.537 *	0.576 *	0.424 *	0.009	0.077
	HOM	0.297 *	0.243 *	0.248 *	0.248 *	0.426 *	0.305 *
	CON	0.554 *	0.488 *	0.569 *	0.211 *	0.159	0.193
	DIS	0.405 *	0.319 *	0.481 *	0.132	0.063	0.093 *
	ENT	0.328 *	0.328 *	0.327 *	0.332 *	0.329 *	0.330 *
	SEC	0.300 *	0.336 *	0.316 *	0.325 *	0.289 *	0.326 *
	COR	0.073	0.079	0.074	0.071	0.000	0.008

Note: * Indicates significant correlation.

**Table 4 plants-14-02948-t004:** Correlation coefficient analysis and maximum value combination between two-dimensional texture index and soil moisture content in different soil layers.

Index	0–20 cm		20–40 cm		40–60 cm	
	Correlation Coefficient	Position	Correlation Coefficient	Position	Correlation Coefficient	Position
ATI	0.643 *	(HOM2, MEA2)	0.615 *	(HOM2, MEA2)	0.576 *	(HOM2, MEA2)
DTI	0.636 *	(MEA3, MEA4)	0.614 *	(MEA3, MEA4)	0.583 *	(MEA3, MEA4)
NDTI	0.537 *	(MEA2, DIS3)	0.598 *	(MEA2, DIS3)	0.559 *	(MEA2, DIS3)
RTI	0.575 *	(CON6, MEA2)	0.520 *	(CON6, MEA2)	0.578 *	(CON6, MEA2)
RDTI	0.496 *	(HOM3, VAR6)	0.467 *	(HOM3, VAR6)	0.472 *	(HOM3, VAR6)
RATI	0.470 *	(VAR6, VAR6)	0.419 *	(VAR6, VAR6)	0.459 *	(VAR6, VAR6)

Note: * Indicates significant correlation.

**Table 5 plants-14-02948-t005:** Correlation coefficient analysis and maximum value combination between the three-dimensional texture index and soil moisture content in different soil layers.

Index	0–20 cm		20–40 cm		40–60 cm	
	R	Position	R	Position	R	Position
BDSI	0.617	(HOM5, CON3, VAR6)	0.598	(VAR4, DIS3, VAR6)	0.584	(HOM5, VAR3, COR6)
DTTI	0.644	(HOM5, HOM2, SEC6)	0.621	(HOM4, DIS1, HOM4)	0.621	(HOM4, DIS1, HOM4)
MSI	0.637	(SEC3, ENT2, DIS1)	0.632	(COR4, VAR6, VAR4)	0.592	(COR5, VAR6, VAR3)
NDTTI	0.622	(SEC6, ENT4, HOM5)	0.612	(ENT2, ENT5, HOM4)	0.567	(VAR2, VAR3, MEA5)

## Data Availability

The original contributions presented in this study are included in the article. Further inquiries can be directed to the corresponding authors.

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
