# Peer review of "Estimating Soil Moisture Content in Winter Wheat in Southern Xinjiang by Fusing UAV Texture Feature with Novel Three-Dimensional Texture Indexes"

_plants, 2025, doi:10.3390/plants14192948_

Round 1

Reviewer 1 Report

Comments and Suggestions for Authors

The main research objective of this article is to accurately estimate the soil moisture content (SMC) of winter wheat in southern Xinjiang by integrating texture features obtained from unmanned aerial vehicles (UAVs) and novel three-dimensional texture indices. The research is conducted under drought conditions to meet the demand for real-time monitoring of soil moisture and ensure crop yields. The paper has certain research significance, but before it can be published, many issues still need to be addressed. My specific opinions are as follows:

1. The parameters of the UAV are not clearly defined, and the band settings and characteristics are not explained at all.
2. What is the basis for choosing the measurement height of the UAV? The pixel resolution and quantity of remote sensing images at different heights vary greatly. How to ensure the applicability of texture feature extraction results at different scales?
3. The paper only mentions a specific growth period of a certain crop. Are the modeling results of this paper applicable to different crop types? How does the author consider the problem of mixed pixels of soil and vegetation?
4. Is it only applicable to the growth period of the crop mentioned in the paper? Are there differences in the results for bare soil or fully vegetated conditions, and is the model still applicable?
5. The essence of this work is to estimate water content by combining texture features of different bands and judging the growth status of vegetation. However, there are many factors affecting crop growth, such as climate, precipitation, farming methods, terrain, soil texture, and physical and chemical properties. How can the author eliminate the influence of these factors? I am skeptical about this.
6. The format of the formulas is not uniform.
7. The reflectance values of the UAV remote sensing images obtained under different flight times and weather conditions in the same band also have significant differences, or the gray levels of the single-band images vary greatly. How to ensure the consistency of texture feature extraction standards in this case? Has the author considered extracting texture features from the results of histogram equalization or binarization of each band?
8. The GLCM texture features have a relatively obvious linear correlation and are placed together with two-dimensional and three-dimensional parameters, resulting in data redundancy.
9. In Tables 1 and 2, the introduction of texture features of different dimensions is too brief. Why were these parameters chosen? What are the standards and significance? What is the calculation process? Even the variables of each parameter are not provided by the author, which is unacceptable. Is it expected for readers to look up the references?
10. The machine learning algorithms mentioned by the author have been widely applied, but the introduction and selection basis of the models, parameter settings, etc. are described too simply. The comparison and analysis of the modeling results of different machine learning algorithms are also not in-depth enough.

Author Response

Reviewer1

Comments and Suggestions for Authors

The main research objective of this article is to accurately estimate the soil moisture content (SMC) of winter wheat in southern Xinjiang by integrating texture features obtained from unmanned aerial vehicles (UAVs) and novel three-dimensional texture indices. The research is conducted under drought conditions to meet the demand for real-time monitoring of soil moisture and ensure crop yields. The paper has certain research significance, but before it can be published, many issues still need to be addressed. My specific opinions are as follows:
Response: Dear reviewer, thank you for your meticulous review and constructive feedback on our manuscript. We appreciate the time and effort you and the first-round reviewers have invested in evaluating our work. We are grateful for your positive assessment of the comprehensive responses and revisions made by the authors. We have now incorporated your comments and suggestions in preparation of the revised manuscript.

1. The parameters of the UAV are not clearly defined, and the band settings and characteristics are not explained at all.

Response: Thank you for pointing this out, we have added the relevant content to the Materials and Methods section.

  1. What is the basis for choosing the measurement height of the UAV? The pixel resolution and quantity of remote sensing images at different heights vary greatly. How to ensure the applicability of texture feature extraction results at different scales?

Response: Thank you for this constructive comment. We have expanded Section 2.2.1 (Multispectral data acquisition and processing) to clarify both the rationale for the 30 m flight altitude and the steps taken to ensure the applicability/consistency of texture features across scale:

Why 30 m AGL?

All missions were flown at 30 m above ground level between 11:00–13:00 local time under clear skies to ensure stable illumination. With our optical setup (DJI M300 RTK + Yusense MS600 Pro), this height yields a native ground sampling distance (GSD) of ≈ 2.2 cm pixel⁻¹, which provides sufficient spatial detail to resolve wheat leaf and stem-cluster structure while retaining efficient area coverage and stable radiometry. This choice follows the common guideline of matching pixel size to the characteristic dimensions of canopy structural elements used for texture analysis (citations added).

Acquisition and preprocessing held constant.

We fixed airspeed (2.5 m s⁻¹) and forward/side overlaps (75%/65%) across flights to ensure geometric and radiometric consistency. Radiometric calibration used a downwelling light sensor (DLS) and white-reference panel. Photogrammetric processing (mosaicking, geometric and radiometric correction) was performed in YuSense Map v2.2.2, and subsequent analyses were conducted on reflectance orthomosaics.

Texture extraction with scale matching.

In ENVI 5.3, we computed eight GLCM-based texture features (MEA, VAR, HOM, CON, DIS, ENT, SEC, COR) on reflectance mosaics using 64 gray levels, a 5 × 5 moving window, and one-pixel offsets at 0°, 45°, 90°, and 135°, with directional averaging to reduce rotational effects. At 30 m AGL (≈ 2.2 cm pixel⁻¹), the 5 × 5 window corresponds to an ≈ 11 cm object-space footprint, which closely matches the spatial scale of wheat leaves and stem communities. To reduce soil–canopy mixing, we applied an NIR-based vegetation mask (threshold τ selected from the image histogram within 0.30–0.40) and refined small gaps/shadows with 3 × 3 morphological opening/closing before computing textures and indices. All quantization, window, and offset parameters were held constant across dates and plots to maintain comparability.

Ensuring applicability across scales.

To avoid scale confounding, all texture features in this study were derived from imagery acquired at the same altitude (30 m) and processed under an identical parameterization. We also note (and have stated in the manuscript) that when multi-altitude imagery must be combined, either (i) resampling imagery to a common GSD or (ii) scaling the texture window to preserve a constant object-space footprint is recommended to maintain the comparability of texture metrics.

These clarifications have been incorporated in Section 2.

  1. The paper only mentions a specific growth period of a certain crop. Are the modeling results of this paper applicable to different crop types? How does the author consider the problem of mixed pixels of soil and vegetation?

Response: Thank you for raising these important points. We have clarified both the scope of applicability and our treatment of soil–vegetation mixing in the revised manuscript:

Scope and transferability.

This study is intentionally scoped to winter wheat at the tillering/jointing period in an arid field setting. We now state explicitly in the Discussion that our conclusions are limited to this crop–growth-stage combination. Extrapolation to other crops, varieties, phenological stages, or sites requires recalibration and independent validation. We outline a transfer protocol in the Discussion: (i) harmonized data acquisition and preprocessing; (ii) reconstruction of candidate 2-D/3-D texture indices; (iii) correlation and collinearity screening tailored to the new context; and (iv) model retraining and independent validation. We also recommend multi-temporal sampling to develop stage-specific or cross-stage models for improved temporal robustness. We note that species/varietal differences in leaf-angle distribution, plant architecture, and canopy closure can change texture expression and the ranking of sensitive band–texture “positions,” further motivating re-selection and re-validation when transferring the workflow. These points have been added to Section 4 (Discussion).

Soil–vegetation mixed pixels.

To minimize soil–canopy mixing, we applied a vegetation mask prior to any texture/index extraction (now detailed in Section 2.2.1). Specifically, we generated an NDVI-based mask and retained pixels with NDVI > τ, where τ was selected from the image histogram within 0.30–0.40. Small gaps and shadows were refined using 3×3 morphological opening/closing, and the final mask was applied before computing GLCM textures and multiband indices on reflectance mosaics. In our plots at the tillering/jointing period, vegetation cover was high, further reducing soil contributions. Additionally, to ensure processing consistency, we fixed flight altitude (30 m; GSD ≈ 2.2 cm), gray-level quantization (64 levels), GLCM window (5×5), and one-pixel offsets at 0°, 45°, 90°, and 135° with directional averaging across all dates and plots. This uniform parameterization helps control scale and processing effects and supports the internal validity of our findings within the stated scope.

These clarifications have been incorporated into Section 2.2.1 (Multispectral data acquisition and processing) and Section 4 (Discussion).

  1. Is it only applicable to the growth period of the crop mentioned in the paper? Are there differences in the results for bare soil or fully vegetated conditions, and is the model still applicable?

Response: Thank you for this valuable question. As clarified in the revised manuscript, our findings are intentionally limited to winter wheat in the jointing period under the arid field conditions of southern Xinjiang. Beyond this scope, we regard our contribution as a generalizable workflow rather than a fixed set of coefficients. In practice, the three-dimensional texture framework treats band–feature “positions” (i.e., combinations of spectral bands and GLCM metrics) as data-driven, context-specific parameters. For other crops, phenological stages, or sites—where canopy architecture (leaf-angle distribution, row spacing, closure), pigment/water status, and soil background differ—the sensitive positions will likely change. This does not alter our study’s logic; it is precisely the strength of the framework, which supports systematic re-selection and fusion of the most informative positions for the new context.

Bare soil or low cover: Prior to any texture/index extraction, we apply a vegetation mask (NDVI-based threshold τ selected from the image histogram within 0.30–0.40, refined with 3×3 morphological opening/closing) on reflectance orthomosaics to minimize soil–canopy mixing. When bare soil predominates, few pixels pass the mask, and texture signals become soil-dominated; models trained at green-up with high cover should not be expected to extrapolate. In such cases, we recommend re-selecting positions, retraining/validating the model on data that explicitly cover low-cover scenes, or stratifying by cover fraction so that stage/cover-consistent models are used.

Full canopy (near closure): At high cover, certain texture measures can saturate and the ranking of sensitive positions can shift. Transfer to fully closed canopies should therefore follow the same screening–modeling–validation pipeline used here, ideally with multi-temporal sampling to derive stage-specific or cross-stage models and to maintain temporal robustness.

Processing safeguards used in this study: To support internal validity within our stated scope, all flights were at 30 m AGL (native GSD ≈ 2.2 cm pixel⁻¹), with overlap 75%/65% and airspeed 2.5 m s⁻¹ held constant. Textures were computed in ENVI on reflectance mosaics using 64 gray levels, a 5×5 window, and one-pixel offsets at 0°, 45°, 90°, 135° (directionally averaged). These fixed parameters reduce scale and processing confounds for the studied stage.

In summary, our results pertain to jointing winter wheat, while our method is portable: apply the same harmonized acquisition/preprocessing → candidate 2D/3D texture construction → correlation/collinearity screening → model training/independent validation workflow, reselect positions appropriate to the target crop/stage/site, and retrain/revalidate to ensure performance and robustness. These clarifications have been added to Section 2.2.1 (Multispectral data acquisition and processing) and Section 4 (Discussion).

  1. The essence of this work is to estimate water content by combining texture features of different bands and judging the growth status of vegetation. However, there are many factors affecting crop growth, such as climate, precipitation, farming methods, terrain, soil texture, and physical and chemical properties. How can the author eliminate the influence of these factors? I am skeptical about this.

Response: Thank you for highlighting this concern. We agree that potential confounders must be explicitly addressed. Our study was designed to isolate soil-moisture (SMC) variation by combining experimental controls with standardized remote-sensing acquisition and preprocessing:

Experimental design controls.

Single, level field with relatively homogeneous sandy soil, minimizing within-site edaphic heterogeneity. Randomized, replicated design with controlled drip-irrigation levels (primary driver of SMC) and fertilizer rates; all plots used the same cultivar, sowing date, seeding rate, and management practices (buffer rows between plots), so management factors did not differ across treatments. Flat topography reduces slope/aspect effects on illumination and imaging geometry.

Standardized remote-sensing acquisition & preprocessing.

Fixed geometry: all flights at 30 m AGL (native GSD ≈ 2.2 cm pixel⁻¹), airspeed 2.5 m s⁻¹, 75%/65% forward/side overlap—held constant across dates. Stable illumination & radiometric calibration: clear-sky windows (11:00–13:00), white-panel plus DLS; identical reflectance-retrieval workflow. Uniform preprocessing: mosaicking/geometric/radiometric correction in YuSense Map v2.2.2; analyses on reflectance orthomosaics. Before computing textures/indices, we applied a vegetation mask (NDVI threshold τ = 0.30–0.40 from image histograms) with 3×3 morphological opening/closing to suppress soil background. Fixed texture parameters: 64 gray levels, 5×5 GLCM window, one-pixel offsets at 0°/45°/90°/135° with directional averaging—kept identical for all plots and dates.

Under these strictly controlled conditions, the associations between multispectral texture features and SMC were established after controlling for weather window, management uniformity, terrain, and soil heterogeneity, i.e., the observed signal reflects SMC differences rather than unrelated factors. We also limit our conclusions to this controlled, field-scale setting (winter wheat at the jointing stage in southern Xinjiang), as stated in the Discussion. For broader applications, we recommend re-applying our workflow with context-specific re-selection of band–texture “positions”, explicit covariate inclusion or stratification, and independent validation across sites, years, and phenological stages. These clarifications have been added to Section 2, and Section 4.

  1. The format of the formulas is not uniform.

Response: Thank you for pointing this out. We have completed a line-by-line check and unified the notation throughout the manuscript:

Replaced all instances of “MEAN” with the standard abbreviation “MEA” in the text, equations, tables, and figure captions.

  1. The reflectance values of the UAV remote sensing images obtained under different flight times and weather conditions in the same band also have significant differences, or the gray levels of the single-band images vary greatly. How to ensure the consistency of texture feature extraction standards in this case? Has the author considered extracting texture features from the results of histogram equalization or binarization of each band?

Response: Thank you for emphasizing cross-mission consistency. Our texture workflow follows quantitative remote sensing practice: we operate strictly in the reflectance domain and standardize both acquisition geometry and GLCM parameters, rather than applying contrast-enhancement transforms that would remap gray-level distributions.

Radiometric consistency.

Each flight used a calibrated white reference panel and a downwelling light sensor (DLS) to retrieve incident irradiance; raw images were converted to surface reflectance via an identical radiometric pipeline before any texture computation. This reflectance-first approach preserves the physical meaning of tone differences across dates and avoids artificial shifts in the gray-level co-occurrence statistics.

Fixed geometry and illumination window.

All missions were flown at 30 m AGL (native GSD ≈ 2.2 cm px⁻¹) with 2.5 m s⁻¹ airspeed and 75%/65% forward/side overlaps, between 11:00–13:00 local time under clear skies, minimizing view-angle and illumination variability across dates.

Uniform preprocessing and texture parameters.

Mosaicking, geometric and radiometric correction were performed in YuSense Map v2.2.2; textures were computed in ENVI 5.3 on reflectance orthomosaics using 64 gray levels, a 5×5 moving window, and 1-pixel offsets at 0°, 45°, 90°, 135° with directional averaging. These parameters were held constant for all plots and both dates to ensure comparability. Prior to texture/indices, we applied an NDVI-based vegetation mask (threshold τ = 0.30–0.40 from image histograms) and refined small gaps/shadows with 3×3 morphological opening/closing to suppress soil background.

Why we do not use histogram equalization or binarization.

Histogram equalization (incl. CLAHE) and binarization remap the gray-level distribution, altering pixel probabilities and co-occurrence matrices; this directly changes the numerical basis of GLCM metrics and breaks cross-date comparability in a quantitative analysis. Because our goal is physically interpretable, cross-mission reflectance-based texture statistics, we avoid any contrast-enhancement step prior to GLCM.

Handling residual cross-date scale differences.

Residual brightness-scale differences after reflectance calibration are addressed at the modeling stage via z-score standardization of predictors within each training fold, which preserves spatial contrast structure while harmonizing units across dates (details in Section 2.3).

These choices (reflectance calibration + fixed acquisition/GLCM settings + vegetation masking + model-stage standardization) provide a consistent and reproducible basis for texture extraction across flights, while avoiding transforms (equalization/binarization) that would distort the statistics we seek to compare. We have added/clarified these points in Section 2 of the revised manuscript.

  1. The GLCM texture features have a relatively obvious linear correlation and are placed together with two-dimensional and three-dimensional parameters, resulting in data redundancy.

Response: Thank you for pointing this out. We agree that multi-band GLCM metrics and their derived 2-D/3-D indices are prone to redundancy and collinearity. Our pipeline addresses this risk both before and within modeling:

Pre-model feature screening (per soil depth, training data only to avoid leakage).

Model-level safeguards:

PLSR projects correlated predictors onto orthogonal latent variables, inherently mitigating collinearity. RF benefits from ensembling and random subspace selection; we report permutation importance (accuracy decrease) rather than split-based counts to reduce bias toward correlated variables. Across models, predictors were z-standardized within each training fold and the same parameters applied to the corresponding validation fold, preserving comparability without leakage.

These procedures, detailed in Sections 2.3–2.4, ensure that only non-redundant, informative predictors enter the models and that remaining correlation is appropriately handled by the learning algorithms.

  1. In Tables 1 and 2, the introduction of texture features of different dimensions is too brief. Why were these parameters chosen? What are the standards and significance? What is the calculation process? Even the variables of each parameter are not provided by the author, which is unacceptable. Is it expected for readers to look up the references?

Response: Thank you for the suggestion. In the revision, Tables 1–2 and their notes now give self-contained explanations: we explicitly define all texture metrics (MEA, VAR, HOM, CON, DIS, ENT, SEC, COR), the six spectral bands (Blue, Green, Red, RE1, RE2, NIR), and the position notation (e.g., HOM6 = homogeneity from NIR). Overall, Tables 1–2 now describe the meaning, selection criteria, and computation steps clearly, without requiring readers to look up additional formulas.

  1. The machine learning algorithms mentioned by the author have been widely applied, but the introduction and selection basis of the models, parameter settings, etc. are described too simply. The comparison and analysis of the modeling results of different machine learning algorithms are also not in-depth enough.

Response: Thank you for this helpful suggestion. Our study is a methodological baseline aimed at testing the feasibility and benefit of combining 3-D texture indices with machine learning for depth-stratified SMC estimation. In the revision, we have expanded Sections 2.3–2.4 and the Discussion to (i) justify model selection, (ii) document hyperparameter tuning in detail, and (iii) deepen cross-model comparisons. These clarifications have been added to Section 2, and Section 4.

Reviewer 2 Report

Comments and Suggestions for Authors

Overall, the manuscript is well written and well designed. Addressing the points below will significantly improve the quality and interpretability of the research.

Comments

  1. Camera specifications

    • Please provide detailed camera specifications, including the central wavelengths of each band. These are essential for proper interpretation of the results.

    • The relationship between plant biophysical/biochemical characteristics and different spectral bands should also be considered.

  2. Figures 2–3

    • It is not clear why, for each band, multiple cells with different values appear on the X and Y axes. This should be clarified.

  3. Figure 4

    • The X-axis has different ranges for each depth. Was it rescaled? For better comparability, it is preferable to keep the same range for all depths.

  4. Figure 5

    • There is no need to present training data; validation results are sufficient.

    • For better interpretation, please add a 1:1 reference line.

    • Despite the high R², which shows clustering of points around the fitted line (but not necessarily the 1:1 line), there is overestimation at low soil moisture contents and underestimation at high soil moisture contents.

    • R² alone is not sufficient to describe accuracy. Consider reporting additional statistics.

    • RMSE is dependent on the magnitude of soil moisture. For instance, the average soil moisture is higher in the 40–60 cm depth compared to the 0–20 cm depth, making direct RMSE comparisons between depths inappropriate.

  5. Discussion

    • The discussion is currently limited. Findings should be compared with previous studies, including UAV-based approaches using other indices or land covers.

    • Please also discuss limitations of the method. For example:

      • Changes in plant species can alter canopy texture.

      • Plant stresses (nutrient deficiencies, diseases) can affect texture in ways unrelated to soil moisture.

      • Texture also varies with phenological growth stages. How can the method be generalized across time and growth stages?

  6. Table 1

    • For better readability, please insert dividing lines between different datasets.

    • The header of the first column should be clearly defined (e.g., Depth).

Author Response

Reviewer2

Comments and Suggestions for Authors

Overall, the manuscript is well written and well designed. Addressing the points below will significantly improve the quality and interpretability of the research.

Response: Dear reviewer, thank you for your meticulous review and constructive feedback on our manuscript. We appreciate the time and effort you and the first-round reviewers have invested in evaluating our work. We are grateful for your positive assessment of the comprehensive responses and revisions made by the authors. We have now incorporated your comments and suggestions in preparation of the revised manuscript.

Comments

  1. Camera specifications

Please provide detailed camera specifications, including the central wavelengths of each band. These are essential for proper interpretation of the results.

The relationship between plant biophysical/biochemical characteristics and different spectral bands should also be considered.

Response: Thank you for the suggestion. We have added the full camera specifications in Section 2.2.1 (Materials and Methods), including center wavelengths and FWHMs for all bands: Blue 450 nm/30 nm, Green 555 nm/27 nm, Red 660 nm/22 nm, Red-edge 720 nm/10 nm and 750 nm/10 nm, and NIR 840 nm/30 nm. To aid interpretation, the revision also summarizes the biophysical/biochemical relevance of each band to crop traits (e.g., red for chlorophyll absorption, red-edge for chlorophyll/structural sensitivity, NIR for canopy structure and water status, blue/green for pigment and backscatter responses). These additions improve transparency and ensure results can be accurately interpreted.

  1. Figures 2–3

It is not clear why, for each band, multiple cells with different values appear on the X and Y axes. This should be clarified.

Response: Thank you for your suggestion. In the revised version, we clarified that the axes in Figures 2-3 list band-texture "positions" and explained in the figure titles: "Each cell shows the Pearson correlation (r) between SMC and a 2-D texture index formed by two "positions" (texture × band, e.g., HOM6, MEA2). The color bar denotes r (−1 to 1). "Position" notation: fᵦ denotes texture f at band b; example: RTI(HOM6, MEA2).

  1. Figure 4

The X-axis has different ranges for each depth. Was it rescaled? For better comparability, it is preferable to keep the same range for all depths.

Response: Thank you for the suggestion. The axes are not scaled or normalized; each panel uses the native units and observed range for that soil layer. We intentionally kept depth-specific x-axis limits because the value scale and error dispersion differ by layer (e.g., similar ?2 but different RMSE/MRE magnitudes). Forcing a single global range compresses some scatters and can mask bias patterns or truncate points, leading to visual misinterpretation. To avoid confusion, the revised caption now states that axis limits are optimized for readability within each depth, and that cross-layer comparisons should rely on standardized metrics (R², RMSE, MRE) reported in the text/tables rather than raw panel scaling.

  1. Figure 5

There is no need to present training data; validation results are sufficient.

For better interpretation, please add a 1:1 reference line.

Despite the high R², which shows clustering of points around the fitted line (but not necessarily the 1:1 line), there is overestimation at low soil moisture contents and underestimation at high soil moisture contents.

R² alone is not sufficient to describe accuracy. Consider reporting additional statistics.

RMSE is dependent on the magnitude of soil moisture. For instance, the average soil moisture is higher in the 40–60 cm depth compared to the 0–20 cm depth, making direct RMSE comparisons between depths inappropriate.

Response: Thank you for these constructive suggestions. We have revised Figure 5 to (i) display only the validation results and (ii) include a 1:1 reference line to aid interpretation. We also acknowledge the observed pattern of slight overestimation at the dry end and underestimation at the wet end, and we have added a brief discussion explaining that this behavior is consistent with regression-to-the-mean and partial saturation of canopy texture signals under moisture extremes. Regarding accuracy metrics, in addition to R² we report RMSE and MRE for the validation set; MRE provides a scale-independent measure of error. Finally, because RMSE is magnitude-dependent, we do not compare RMSE across soil depths; we now state this explicitly and base cross-depth comparisons on MRE.

  1. Discussion

The discussion is currently limited. Findings should be compared with previous studies, including UAV-based approaches using other indices or land covers.

Please also discuss limitations of the method. For example:

Changes in plant species can alter canopy texture.

Plant stresses (nutrient deficiencies, diseases) can affect texture in ways unrelated to soil moisture.

Texture also varies with phenological growth stages. How can the method be generalized across time and growth stages?

Response: Thank you for this constructive guidance. In the revised Discussion, we (i) add a focused comparison with recent UAV studies across three strands—spectral-only models, spectral + 2-D texture, and multi-modal settings (e.g., thermal fusion; different crops/land covers)—to position our accuracy and to highlight the incremental benefit of 3-D texture fusion under field conditions similar to ours; and (ii) articulate method limitations and paths to generalization. Specifically, we note that species/varietal differences (leaf-angle distribution, architecture, canopy closure) can shift texture expression and the ranking of sensitive band–texture “positions”; non-water stresses (nutrient deficiency, disease) may induce texture changes unrelated to SMC; and phenological dynamics alter texture with changing bare-soil fraction, LAI, and structural reorganization. To address these issues, we recommend (and outline) a transferable workflow: harmonized acquisition/preprocessing, re-construction and re-selection of positions for the target crop/stage/site, multi-temporal sampling with stage-specific or cross-stage models (phenology covariates or temporal normalization), and independent validation (e.g., leave-one-site-out, time-blocked CV). We also suggest including stress covariates (chlorophyll/thermal indicators, pest–disease records) or stratifying by cover/stress to disentangle water-related from non-water effects. These additions strengthen interpretability and scope without changing the study’s conclusions. Relevant text appears in the Discussion.

  1. Table 1

For better readability, please insert dividing lines between different datasets.

The header of the first column should be clearly defined (e.g., Depth).

Response: Thank you for pointing that out; we have made the changes as per your suggestion.

Round 2

Reviewer 1 Report

Comments and Suggestions for Authors

The reviewer would like to say that the new version has been well improved. In my point, it is quite ready for acceptance. 

Reviewer 2 Report

Comments and Suggestions for Authors

My comments were addressed.